# Vesicle shape transformations driven by confined active filaments

Matthew S. E. Peterson [1], Aparna Baskaran [1✉] & Michael F. Hagan [1✉]

In active matter systems, deformable boundaries provide a mechanism to organize internal active stresses. To study a minimal model of such a system, we perform particle-based simulations of an elastic vesicle containing a collection of polar active filaments. The interplay between the active stress organization due to interparticle interactions and that due to the deformability of the confinement leads to a variety of filament spatiotemporal organizations that have not been observed in bulk systems or under rigid confinement, including highly-aligned rings and caps. In turn, these filament assemblies drive dramatic and tunable transformations of the vesicle shape and its dynamics. We present simple scaling models that reveal the mechanisms underlying these emergent behaviors and yield design principles for engineering active materials with targeted shape dynamics.

---

[1] Martin A. Fisher School of Physics, Brandeis University, Waltham, MA 02453, United States. ✉email: aparna@brandeis.edu; hagan@brandeis.edu

Active matter encompasses systems whose microscopic constituents consume energy at the particle scale to produce forces and motion. Novel macroscale phenomena emerge in these systems when these forces collectively organize into mesoscale 'active stresses'. For example, many biological functions, such as cytoplasmic streaming, morphogenesis, and cell migration, are driven by active stresses that emerge from active components confined within a cell[1–8]. Understanding physical mechanisms that underlie these functions is a key goal of cellular biophysics. From a technological perspective, harnessing active stresses to drive particular emergent behaviors could enable a new class of materials with life-like properties that would be impossible in traditional equilibrium materials. However, rationally designing active constituents to generate a desired emergent behavior requires identifying the principles that govern organization of mesoscopic active stresses.

In particular, the membrane protrusions (filopodia and lamellipodia) that underlie these processes are driven by bundled actin filaments undergoing polymerization; at the same time, actin-bundling is enhanced by funneling of the filaments by the resulting membrane curvature[9]. Achieving such capabilities in minimal bio-derived experimental systems (e.g. refs. [9–12]) is essential to engineer controlled shape transformations. There have been a number of numerical studies forcused on specific aspects of the actin cortex organization driving these phenomena[13–16]. However, given the complexity of active matter systems, it is important to develop and study minimal models, which focus on specific mechanical aspects of the complex emergent behavior. More broadly, such studies will guide the design of soft robotics, artificial cells, or other advanced materials that mimic the capabilities of living organisms.

The field of active matter has identified two key mechanisms that provide control over active stress organization: (1) anisotropic interactions between active components that realign forces, and (2) confining boundaries. These mechanisms fundamentally differ from the effects of internal stresses and boundaries in equilibrium systems[17]. Due to their persistent motion, aligned active particles can generate system-spanning net forces or flows. For example, interactions between self-propelled particles that drive interparticle alignment result in bands or flocks[18,19], and changing the length and stiffness of active polymers leads to dramatic reorganization of active stresses[20–25]. While boundary effects are typically short-ranged in an equilibrium system, confining an active system can redirect the hierarchical organization of its internal active stresses and thus qualitatively change its macroscopic emergent behavior. For example, confining active particles leads to system-spanning effects such as spontaneous flow[26–33]. Furthermore, deformable confining boundaries enable non-equilibrium boundary fluctuations[12,34–38], including elongated tendrils and bolas[35] and budding[39]. The latter results suggest that flexibility is a key characteristic of a confining boundary, as it allows shape transformations, sensing and response to environmental cues.

Despite the important insights from these studies, relatively little is known about the behaviors that may arise when these two active stress organization modes are coupled—that is, enclosing *anisotropic* active components within *deformable* boundaries. In particular, most existing theoretical and computational studies have focused on rigid boundaries[40–43], isotropic active particles[34–36,44–48], or have been in 2D[22,49,50]. More closely related to our work are simulation studies of droplets containing active material that show tantalizingly life-like behaviors such as motility and division[51–54]. These elegant studies highlight the importance of understanding the types of emergent behaviors that arise when active matter and deformable boundaries are combined. However, the continuum hydrodynamic theories employed in these works require key assumptions about the nature of particle organization and particle-membrane interactions.

Here, we describe a particle-based computational study of a minimal model that combines active interparticle alignment interactions and deformable confinement, in which the form of particle assemblies and their interactions with the membrane are an emergent property of the dynamics. In particular, we use Langevin dynamics simulations of polar self-propelled semiflexible filaments confined within 3D elastic vesicles. Rather than directly describing the cytoskeleton as in recent computational studies[9,13–16], we seek a minimal mechanical model that can describe shape transformations in vesicles deformed by active stresses. Thus, we consider active filament propulsion in the overdamped limit without long-ranged hydrodynamic coupling. Our results thereby identify a generic route to control self-organizing active stresses by enclosing active components with anisotropic shapes and/or internal degrees of freedom within deformable confining boundaries.

The simulations show that the interplay between these two methods of active stress organization leads to a positive feedback, in which active forces drive boundary deformation while passive stresses from the boundary re-align and reinforce self-organization of the internal active stresses. This leads to a rich variety of steady-state behaviors that have not been observed in bulk systems or under rigid confinement, including highly aligned rings, and caps that have tunable self-limited sizes, number, and symmetry. Each filament organization drives a characteristic large-scale vesicle shape transformation that can be selected by varying parameters such as filament length, density, and flexibility. We also present simple scaling analyses that reveal how the feedback between vesicle geometry and filament organization drives and stabilizes these emergent behaviors. The applicability of these scaling arguments suggests that these behaviors arise generically due to feedback between vesicle elasticity and active filament organization, independent of the specific model.

## Results

To discover the steady-state conformations that arise due to coupling between active propulsion and elasticity, we have performed simulations over a wide range of control parameters which we present in nondimensional form as: the volume fraction of filaments in the vesicle $\phi \in [0.01, 0.4]$, filament aspect ratio $a = 1 + (M - 1)(b_{\mathrm{fil}}/\sigma) \in [3, 25.5]$ (with $b_{\mathrm{fil}}$ and $\sigma$ the mean bond length and monomer diameter, respectively), Péclet number $\mathrm{Pe} = f_{\mathrm{a}}\sigma/k_{\mathrm{B}}T \in [0, 10]$ (with $f_{\mathrm{a}}$ the active propulsion force, and $T$ the system temperature), filament rigidity $\chi_{\mathrm{ves}} = \kappa_{\mathrm{fil}}/k_{\mathrm{B}}T \in [10^2, 10^4]$ (with $\kappa_{\mathrm{fil}}$ the filament bending modulus), and vesicle rigidity $\chi_{\mathrm{ves}} = \kappa_{\mathrm{ves}}/k_{\mathrm{B}}T \in [10^2, 10^4]$ (with $\kappa_{\mathrm{ves}}$ the vesicle bending modulus). Additional simulation details can be found in the Methods section.

Figure 1 shows the steady-states as a function of filament volume fraction and aspect ratio for moderate activity $\mathrm{Pe} = 8$. At this activity and vesicle size, for aspect ratios $a \gtrsim (24 R_{\mathrm{ves}}/\sigma \mathrm{Pe})^{1/3} \approx 4.2$ the system is in the *strong confinement limit*: because the persistence length of the filament center-of-mass motion, $l_{\mathrm{p}}^{\mathrm{COM}} = \sigma a^3 \mathrm{Pe}/12$, is larger than the vesicle size, $l_{\mathrm{p}}^{\mathrm{COM}} > 2R_{\mathrm{ves}}$, most filaments are found on the vesicle surface at all times[55] (see Supplementary Note 1, see Supplemental Material for model details and additional figures).

Under these conditions we can classify the steady-state vesicle conformations into several categories: *(I)* spherical, *(II)* oblate, *(III)* polar-prolate, *(IV)* apolar-prolate, and *(V)* polyhedral. These vesicle configurations are tightly coupled to the spatiotemporal organization of the filaments within, as follows.

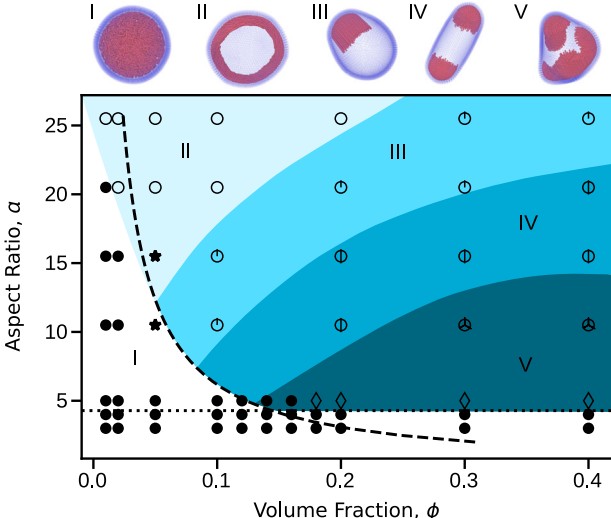

**Fig. 1 Steady-state configurations as a function of filament volume fraction $\phi$ and aspect ratio $a$.** Snapshots illustrating steady-state configurations of the vesicle and enclosed active filaments as a function of filament aspect ratio $a$ and initial volume fraction $\phi$. See Supplementary Movie 1 for animations of the corresponding simulations (See Supplemental Material for model details and additional figures). The marked regions of parameter space indicate the typical vesicle conformation: (I) spherical, (II) oblate, (III) polar-prolate, (IV) apolar-prolate, and (V) polyhedral. The symbols associate the conformation with the internal filament organization: homogeneous throughout the bulk or on the surface, with no vesicle deformation (●); transient clusters and/or bands, with oblate vesicle shapes (★); stable polar rings (⊙); stable caps (circle around a number of intersecting lines equal to the median number of caps); and dynamic caps (◇). The dashed line shows the transition to aligned states predicted from the competition between the characteristic collision and reorientation timescales ($\phi = (\pi/4)^2/a$) described in the text, and the horizontal dotted line indicates the approximate threshold aspect ratio for the filaments to be in the strong confinement limit. Other parameters are the Péclet number Pe = 8, filament bending modulus $\chi_{fil} = 10^4$ and vesicle rigidity $\chi_{ves} = 5 \times 10^3$.

*(I)*: Spherical vesicle shapes arise at low filament volume fractions and aspect ratios. Under these conditions, filament-filament collisions are rare and inter-filament aligning forces are weak[56–62]. Thus, filament positions and orientations are homogeneous (throughout the vesicle interior below strong confinement, or on the vesicle surface above strong confinement), leading to little deformation of the vesicle.

*(II)*: For low volume fraction but high aspect ratios, such that the filament length $L = a\sigma$ is comparable to the unperturbed vesicle radius, $L \sim R_{ves}$, the vesicle deforms into oblate spheroid conformations. This transition is driven by the filaments organizing into a stable polar band in which all filaments move in the same direction, resulting in deformations of the vesicle along a geodesic. This filament arrangement closely resembles the polar bands observed on the surface of rigid spheres for active particles with polar propulsion and polar interparticle alignment interactions[46], which arise due to topological requirements for a surface-constrained polarization field[42]. However, note that such polar bands would be unstable to fluctuations that are transverse to the polar orientation of the band in our system if the confining geometry was a rigid sphere because the filament-filament interactions in our system are nematic (head-tail symmetric)[63,64]. The finite deformability of the vesicle is essential to stabilize this configuration—active forces due to the polar band force the vesicle into an oblate shape, which in turn provides a

restoring force to stabilize transverse fluctuations to filament alignment within the band. In support of this conclusion, simulations on infinitely rigid vesicles did not exhibit stable polar bands (see below). Thus, this configuration provides a concrete example of how feedback between passive stresses and self-organization of active stresses can generate steady states that would be otherwise disallowed by symmetry. To further elucidate this interplay between passive and active stresses, we present a theoretical analysis in Supplementary Note 6 and Supplementary Figure 6 that relates the extent of vesicle deformation to the forces exerted by the filaments within the band.

*(III-V)*: For intermediate volume fractions and aspect ratios, the vesicle deforms into a prolate spheroid. These prolate vesicle conformations can be further classified by the net alignment of the enclosed filaments as either polar *(III)* or apolar *(IV)*. We note that states with net polarity can exhibit center-of-mass motion, but more comprehensive models that account for momentum conservation would be important to study such effects. Therefore, in what follows we focus on the internal motions and shape transformations of the vesicle. Further increasing the volume fraction or decreasing the aspect ratio leads to polyhedral conformations, *(V)*. States *(III-V)* all result from filaments assembling into crystalline caps in which the rods are highly aligned and perpendicular to the vesicle surface. Interestingly, the caps are 'self-limited' in that their typical size decreases with decreasing aspect ratio, but is roughly independent of the total number of filaments $N_{fil}$ in the vesicle. Increasing $N_{fil}$ at fixed aspect ratio increases the number of caps; we observe up to 12 caps for the finite vesicle size that we consider (see below). Further, caps drive local curvature of the vesicle, leading to elasticity-mediated cap-cap repulsions, which favor symmetric arrangements of caps. Thus, the vesicle morphology can be sensitively tuned by controlling filament aspect ratio and density to achieve a specific number of caps. The polar-prolate *(III)*, apolar-prolate *(IV)*, and polyhedral states *(V)* respectively have 1, 2, and ≥3 caps. For states with small numbers (1-3) of caps, the filament organization is highly stable once the system reaches steady state. However, for enough caps in the vesicle (typically more than 3), the caps can become motile, and collide with, merge with, and split from other caps (see below). Note that the formation of these caps is qualitatively similar to the buds that were observed in simulations of surface-bound proteins, which have attractive interaction-induced aggregation and exert forces in the normal direction of a highly deformable membrane[39]. In contrast, in our system the filaments interact only through excluded volume and are not constrained to the membrane; thus, caps are an emergent property arising from activity, excluded volume, and passive forces from the membrane.

**Mechanisms underlying stress organization and deformation.** To understand how these conformations are governed by the interplay between propulsion-induced aligning forces, vesicle deformability, and vesicle curvature, we develop simple scaling estimates for the timescales and forces that govern filament alignment and interactions with the vesicle. First, we consider the transition between undeformed spherical vesicle states characterized by unaligned or weakly aligned filaments as in state *(I)*, to the highly deformed oblate, prolate, and polyhedral vesicle shapes of states *(II-V)*. Our simulations demonstrate that such significant vesicle shape deformations occur when filament-filament interactions mediate the organization of ordered structures either in the plane of the vesicle or orthogonal to it. (The ability of collisions of self-propelled particles on a surface to drive formation of smectic layers is supported by a recent observation in bacterial colonies growing on flat surfaces, in which bacteria

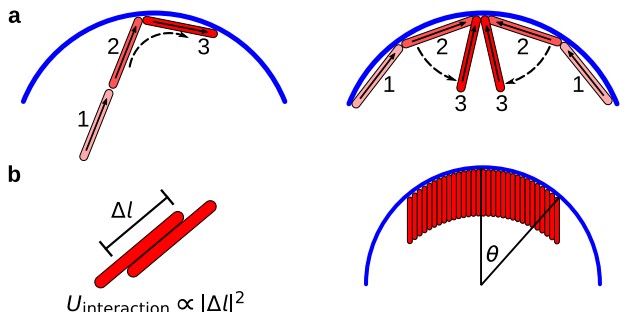

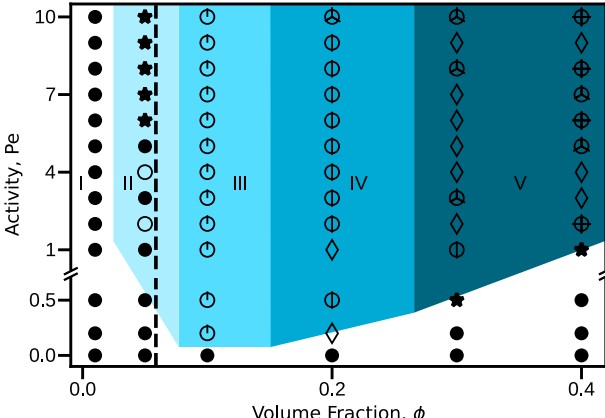

**Fig. 2 Mechanisms of filament assembly into rings and caps. a** The onset of ring and cap formation is determined by a competition of timescales: the timescale associated with rotations parallel to the vesicle (left) and the timescale associated with collisions that tend to orient the filaments perpendicular to the vesicle (right). **b** Schematic of the theory for the number of caps (Eqs. (1) and (2)). We assume an activity-induced effective attractive interaction that is quadratic in the rod-rod contact length $\Delta l$ (left). The cap is assumed circular, with size parameterized by the angle $\theta$ between the cap center and edge. Vesicle curvature leads to a shearing of rods within the cap (right).

**Fig. 3 Steady-state configurations as a function of volume fraction $\phi$ and Péclet number Pe.** The marked regions are defined as in Fig. 1. The dashed line shows the transition to aligned states predicted by the timescale competition, which is independent of Pe. Other parameters are $a = 10.5$, $\chi_{\text{fil}} = 10^4$, and $\chi_{\text{ves}} = 5 \times 10^3$. Note that the figure has a change in scale on the activity axis for Pe < 1. See Supplementary Movie 2 for corresponding animations.

form 'rosettes' with the rod-like bacteria oriented perpendicular to the surface[65]. However, in contrast to the self-limited caps in our system, the bacterial rosettes do not exhibit a preferred size because they are on a flat boundary). In the following discussion we assume the strong confinement limit; in particular, we assume that the timescale governing the rotational diffusion of the filaments, $\tau_{\text{corr}}$, is long compared to the other relevant timescales.

**The onset of filament assembly**. The onset of this transition can be understood by considering a competition between two characteristic timescales that respectively govern collision-induced filament-vesicle alignment and filament-filament alignment (see Fig. 2). Filament-vesicle collisions, which tend to reorient filaments parallel to the surface[66,67], have a characteristic timescale $\tau_{\text{rot}} \sim L/v_a$ with $v_a = \text{Pe}v_0$ the filament self-propulsion velocity and $v_0 = k_B T/m\gamma\sigma$ a characteristic velocity of the system. We can estimate the timescale for filament interactions by considering filament-filament pairwise collisions whose timescale is given by $\tau_{\text{coll}} \sim \sigma/v_a\phi$ (see Supplementary Note 2, see Supplemental Material for model details and additional figures). Thus, deformed vesicle states will arise when $\tau_{\text{coll}} < \tau_{\text{rot}}$ or equivalently $a\phi > c$, where $c \cong (\pi/4)^2$ is independent of activity and filament length (See Supplemental Material for model details and additional figures). This defines a boundary separating highly deformed states of the vesicle from the undeformed spherical states (the dashed line in Fig. 1).

Notably, the active force drops out of this argument because both collision and reorientation times are $\propto$ Pe. Thus, the theory predicts that the emergence of deformed vesicle states is independent of activity of the enclosed filaments (above a threshold activity). As a test of this prediction, Fig. 3 shows the steady-states as a function of $\phi$ and Pe for fixed aspect ratio $a = 10.5$. Indeed, formation of large deformations does not depend on activity, with non-spherical shapes forming for $\phi \geq c/a \approx 0.06$ (as predicted by the above timescale argument) for all Pe > 0 that we considered.

This simple theoretical picture gives a predictive principle, in terms of properties of the active filaments, for when vesicle shape transformations occur. However, the theory assumes the strong activity, long filament limit and thus neglects thermal noise. Below a threshold activity (Pe $\lesssim$ 1) the vesicle will not deform because filament organization is destroyed by thermal

fluctuations. Also, cap formation (and thus vesicle shape transformation) does not occur when the filaments are below the strong confinement limit discussed earlier ($a \lesssim 4.2$ for the parameters of Fig. 1, shown as a dotted line). This observation is consistent with the assumption made during the preceding analysis, that the rotational diffusion timescale of the filaments, $\tau_{\text{corr}}$, is much larger than both $\tau_{\text{rot}}$ and $\tau_{\text{coll}}$.

**Cap morphologies**. We can derive further insight into shape transformations by considering the system in the strongly deformed regime with polyhedral shapes. The defining characteristic underlying these states is filament assembly into well-ordered caps. Most cap states are relatively static, with occasional association/dissociation of individual rods (See Supplementary Movie 4), except for the parameters that lead to the highly dynamic, reconfiguring caps discussed below. In a static steady state, the active and elastic forces must balance. In particular, the dense crystalline nature of caps arises because the active force and the presence of the vesicle surface leads to an effective attractive interaction between nearby filaments. This attraction drives radial growth of a cap, since filaments on the cap periphery have fewer neighbors, leading to an effective interfacial tension. This effect is both reinforced by and competes with vesicle elasticity. The active force of small caps drives vesicle deformations whose local curvature enhances effective filament-filament attractions. However, as the cap grows in radius, vesicle curvature drives an effective shear of filaments (see Fig. 2b) that reduces rod-rod overlaps and thus opposes the active force.

We describe this competition by constructing an effective 'free energy' whose gradients correspond to the active and passive forces (Fig. 2b). Since the active force favors rods to align in a smectic layer, the shear due to vesicle curvature imposes an 'energy' cost of $U_{\text{shear}}(\theta) = n_{\text{cap}}2\pi R_{\text{ves}}^2 \frac{G}{2}[\cos\theta + \sec\theta - 2]$, with $\theta$ the angle subtended by the cap on the vesicle surface, $n_{\text{cap}}$ the number of caps, and a 'shear modulus' $G \sim$ Pe (but independent of $L_{\text{rod}}$) (See Supplemental Material for model details and additional figures). In the strongly deformed region the caps are roughly circular, so the interfacial energy is given by $U_{\text{int}}(\theta) = n_{\text{cap}}2\pi R_{\text{ves}}\lambda \sin\theta$, with 'interfacial tension' $\lambda \sim L_{\text{rod}}$Pe accounting for the diminished interactions at the cap boundary. This results in a free energy as a function of cap size (See Supplemental Material for model details

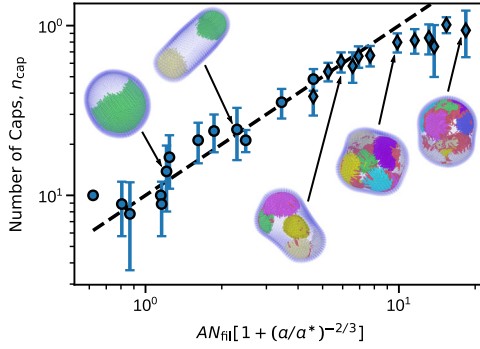

**Fig. 4 The number of caps formed in simulations compared to theory.**
Symbols represent mean numbers of caps measured from simulations, with diamonds indicating dynamic cap states and the error bars showing the standard deviation. The dashed line is the expectation from theory given by Eq. (2). Note that the number of caps in the simulation results is likely under-counted for the dynamic states due to the caps' motility. The simulation data is taken from Fig. 1 and consists of all states that form rings or caps. For the theory, $A = 1/2\pi\rho R_{ves}^2$ and $a^* \approx 130$ (from fitting to data points with $n_{cap} \leq 7$; points with more caps were not included in the fit due to the unreliable counting). Active filaments are colored by which cap they belong to for visual clarity. Note that both filament aspect ratio and volume fraction are changed between snapshots.

and additional figures):

$$f(\theta) = \frac{1}{1 - \cos\theta}\left[\frac{1}{2}(\cos\theta + \sec\theta - 2) + \zeta\sin\theta\right] \quad (1)$$

where $\zeta = G/\lambda R_{ves} \sim L/R_{ves}$ is given by the balance between the effective interfacial tension and shear modulus, and should be linear in filament length but roughly independent of Pe since both of these effects are driven by activity.

Minimizing this per-filament free energy yields an optimal $\theta$[68,69] corresponding to the self-limited cap size. Assuming that we are well above the onset of cap formation so that essentially all filaments are in caps,

$$n_{cap} \propto N_{fil}[1 + (a/a^*)^{-2/3}] \quad (2)$$

where $a^* \propto R_{ves}/\sigma$ is an adjustable parameter that depends on the local vesicle curvature at the cap, which results from a balance between active forces from the filaments and passive forces from vesicle bending (see Supplementary Note 3). Thus $a^*$ may depend on the moduli of the vesicle and filaments as well as activity in some limits. This expression holds provided $a \ll a^*$. For the data in Fig. 1, we obtain $a^* \approx 130$, leading to the dashed line shown in Fig. 4.

Except for states with many ($n_{cap} \gtrsim 7$) motile caps, there is close agreement between the observed and predicted $n_{cap}$. Above this threshold our cap-counting algorithm likely under counts $n_{cap}$, since different caps are often adjacent and interacting. Further, the prediction of Eqs. (1) and (2) that the self-limited cap size is independent of activity is consistent with observations at different Pe (see Fig. 3). The motile cap states appear to arise when the curved vesicle geometry forces interactions between the inward-facing ends of adjacent caps. Such interactions occur above a threshold number and aspect ratio of filaments, given by $N_{fil} \gtrsim C(1 - a\sigma/R_{ves})^2$, where $C \propto R_{ves}^2$ is a constant (see Supplementary Note 4, see Supplemental Material for model details and additional figures).

We note that the geometric factors governing the self-limited cap size parallel those in a recently studied *equilibrium* system of rigid filaments end-adsorbed onto a rigid spherical nanoparticle, which self-assemble due to direct pairwise inter-filament attractions[69]. However, in the present system, the effective filament-filament interactions and vesicle geometry are many-body and emergent in

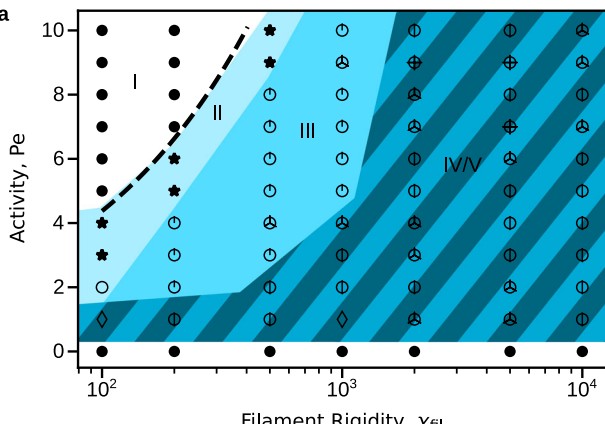

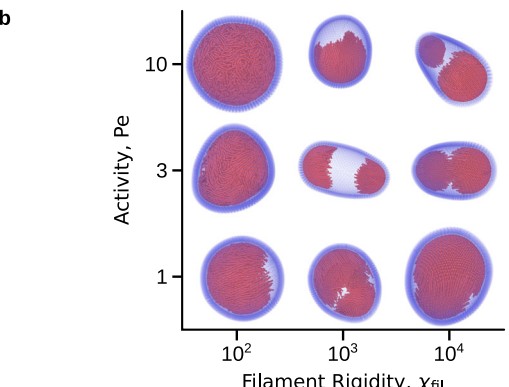

**Fig. 5 Increasing activity decreases the effective filament bend modulus, causing the system to leave the strong confinement limit. a** Vesicle conformations and filament organizations as a function of filament rigidity $\chi_{fil}$ and active force strength Pe, for fixed volume fraction $\phi = 0.2$ and filament aspect ratio $a = 10.5$. For a given filament stiffness, increasing activity reduces the number of caps until an upper-threshold activity value $Pe_{SC}$, beyond which the system transitions into an undeformed state. As described in Supplementary Note 5, this transition occurs because activity renormalizes the filament bending modulus to smaller values[70], thus reducing filament alignment interactions and causing the system to leave the strong confinement limit. The dashed line shows the prediction for $Pe_{SC}$ given by Supplementary Eq. 35. Note that there is no adjustable parameter. In the rigid rod limit ($\chi_{fil} > 10^3$) all nonzero active force values that we simulated led to cap formation. **b** Selected snapshots of states shown in (**a**). Animations of these states can be found in Supplementary Movie 3.

that they arise due to feedback between nonequilibrium active forces and vesicle deformations.

**Effect of filament and vesicle rigidity.** Thus far, we have focused on the interplay between activity and vesicle deformability by performing simulations in the limit of rigid rods, $\chi_{fil} = 10^4$, and high (but finite) vesicle rigidity $\chi_{ves} = 5 \times 10^3$. We now briefly discuss the effect of allowing for finite filament and vesicle flexibility.

Figure 5 shows the vesicle conformation and filament organization states as a function of filament bending modulus and activity, for fixed filament volume fraction $\phi = 0.2$. We see that for finite filament flexibility, the transition to aligned ring and cap states is suppressed *above* a threshold activity, which decreases with decreasing $\chi_{fil}$.

This result can be understood as follows. On generic grounds, decreasing the filament rigidity will reduce the tendency for

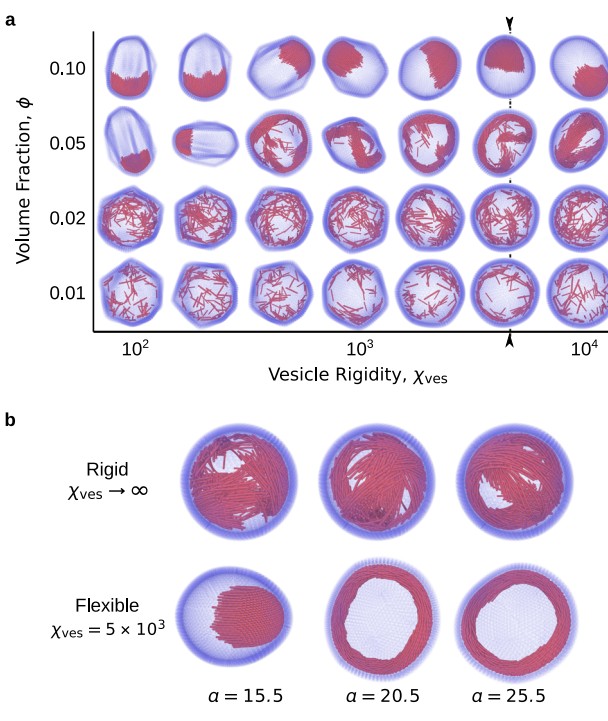

**Fig. 6 Vesicle deformability is critical for stable ring and cap formation. a** Simulation snapshots illustrating vesicle conformations and filament organizations as a function of filament volume fraction and vesicle rigidity. As the vesicle rigidity is reduced below a critical value (corresponding to the critical Föppl–von Kármán number FvK ≈ 154[75], indicated by the dashed line), the vesicle undergoes a buckling transition leading to the formation of facets. While we observe most of the same classes of filament self-organization in faceted and round vesicles, polar bands trace a dynamic path between vertices in faceted vesicles, while they trace a stable geodesic in round vesicles. Animations of these states can be found in Supplementary Movie 5. **b** Comparisons between flexible ($\chi_{ves} = 5 \times 10^4$) and rigid ($\chi_{ves} \to \infty$) vesicles as a function of filament aspect ratio, with other parameters set to Pe = 8, $\phi$ = 0.1, and $\chi_{fil} = 10^4$. In contrast to flexible vesicles, rigid vesicles do not allow for the formation of stable caps or rings. When polar rings do form in rigid vesicles, they are transient---continuously breaking and reforming over the course of the trajectory. Animations of this comparison can be found in Supplementary Movie 6.

filaments to align and thus impede the formation of aligned rings and caps. For filament stiffness values well below the rigid rod limit, the process by which caps and rings form is more complicated than considered previously. The upper-threshold activity for filament organization can be, at least in part, explained by the observation that activity renormalizes filament rigidity to smaller values according to $\chi_{fil}^{eff} \cong \chi_{fil}/(1 + Pe^2)$[70]. Interactions between flexible active agents are such that the active energy preferentially dissipates into bend modes, effectively increasing filament flexibility and therefore suppressing filament alignment. In particular, the upper-threshold activity corresponds to the point when the activity-renormalized flexibility of filaments causes the system to leave the strong confinement limit. This occurs for Pe $\gtrsim C\chi_{fil}^{3/5}$, where $C = (24R_{ves}/\sigma)^{-1/5}$ (see Supplementary Note 5 for details), which is shown as the dashed line in Fig. 5. Accounting for this activity-renormalized filament flexibility can also be used in Eq. (2) to explain the reduction in the number of caps as activity is increased at a fixed filament rigidity.

Figure 6 a shows the conformations obtained by varying the vesicle rigidity $\chi_{ves}$ and filament volume fraction $\phi$, while fixing

the filament rigidity $\chi_{fil} = 10^4$ and activity Pe = 8. The most striking effect of reducing the vesicle rigidity is that it drives a faceting transition when the Föppl–von Kármán number, FvK = $YR_{ves}^2/\kappa$ where $Y = 2k_{ves}/\sqrt{3}$ is the Young's modulus of the vesicle and $\kappa = \sqrt{3}k_B T\chi_{ves}/2$ is the bending modulus[71–73], is increased above a critical value, FvK $\gtrsim$ 154. This is an equilibrium property of an elastic vesicle, independent of the active filaments[74]. Our results indicate that faceting does not qualitatively change the formation of caps, but that caps form at slightly lower filament volume fraction for reduced vesicle bending modulus. This could be anticipated from the theoretical arguments described above, since reducing the bending modulus allows filaments' active forces to further deform the vesicle, leading to a smaller local radius of curvature in the vicinity of a cap. More interestingly, the facets appear to destabilize the polar bands and rings. For round vesicles (with bending modulus such that FvK < 154), a stable ring forms along a geodesic. In contrast, in faceted vesicles at the same activity and filament volume fraction, rings or bands tend to form paths that connect facet vertices. The bending of the ring path imposed by the facet connectivity destabilizes the ring, causing it to transiently break and reform (similar to the transient band state described above). This behavior suggests that it will be interesting to explore the possibility of coupling between vesicle faceting and filament organization in a future work.

Figure 6b compares configurations observed with a flexible vesicle ($\chi_{ves} = 5 \times 10^3$) and a rigid vesicle ($\chi_{ves} \to \infty$) for $a \in [15.5, 25.5]$, $\phi = 0.10$, Pe = 8, and $\chi_{fil} = 10^4$. While the flexible vesicle exhibits stable polar rings and single caps at these parameters (Fig. 1), the rigid vesicle system is unable to form the single-cap state, and only exhibits transient polar rings, which continuously break apart and reform as the simulation progresses. These results emphasize the importance of the feedback between active stress organization and vesicle deformation, which allows for stable states that are otherwise inaccessible under rigid confinement.

## Discussion

This work demonstrates that confining active filaments within a deformable vesicle leads to multiple transformations of the vesicle shape and motility, which can be precisely tuned by control parameters. The feedback enabled by coupling deformable boundaries with anisotropic particles significantly enriches the available modes of self-organization. While the self-limited caps are the most striking class of such behaviors, the stable polar bands for particles with nematic interactions provides a clear example of how boundary deformations can stabilize novel states. Notably, both of these classes of behaviors arise due to a spontaneous symmetry breaking of the initially spherical boundary.

While we emphasize that our minimal model is not intended to represent specific biomolecular systems, it is interesting to note that the lateral coalescence of filaments and the resulting membrane protrusions of cap states in our system bear resemblance to the organization of cortical actin within lamellipodia in biological cells[9,13–16]. Although the filament activity in our model has the same polar symmetry as in the actin cortex, there are also key differences. In the cortex, activity derives from asymmetric polymerization of actin rather than filament self-propulsion, and bundling of actin filaments is driven by cross-linking proteins and branching agents, in addition to forces deriving from membrane curvature. Since we do not include lateral attractive interactions between filaments in our minimal model, the lateral coalescence of filaments within caps is driven entirely by their persistent motion and membrane curvature. The fact that we observe similar filament arrangements and coupling to the membrane in a

minimal model thus suggests that this may be a generic mechanism for driving membrane protrusions.

These results have implications for future experiments on active materials constructed from anisotropic particles confined within deformable boundaries. In particular, the transitions can be controlled by tuning parameters that are readily accessible in experiments—filament length, flexibility, and volume fraction. In contrast, activity is a complicated function of motor properties and ATP in bio-derived systems[75,76]. Thus, our computational results suggest strategies to engineer active vesicles with designable shapes and dynamics, and other capabilities resembling those of living cells. Furthermore, our theoretical analysis identifies the mechanisms that underlie these emergent morphologies by revealing how filament-filament interactions and vesicle deformations couple to spatiotemporally organize stress. This provides a model-independent roadmap for exploring additional classes of emergent functionalities in parameter regimes beyond the scope of the present work, including highly deformable fluidized vesicles and other symmetries of activity.

## Methods

We consider a minimal model to study stress organization at a deformable boundary through contact interactions. We simulate a system of $N_{fil}$ active filaments confined within an elastic vesicle, which has radius $R_{ves}$ in its undeformed state. We represent active filaments using the model in Joshi, et al.[70]—modified so that the active forcing is polar rather than nematic—in which each filament is a nearly-inextensible, semiflexible chain of $M$ beads of diameter $\sigma$[24]. Bonded beads interact through an expanded FENE potential[77], while non-bonded beads interact through a purely repulsive expanded Weeks-Chandler-Andersen (eWCA) potential[78] with strength $\epsilon$. The equilibrium bond length is set to $b_{fil} = \sigma/2$ to minimize surface roughness between interacting filaments, thereby preventing filaments from interlocking at high density[23,25,79–81]. The filaments are made semiflexible with bending rigidity $\kappa_{fil}$ through a harmonic angle potential applied to each set of three consecutive beads along the chain.

To focus on the interplay between active stresses in deformable boundaries, our model is not intended to describe any specific biofilament system and incorporates activity in a minimal manner—a polar active force of magnitude $f_a$ acts on each bead, in a direction tangent to the filament and toward the filament head. This active force is meant to account for propulsive forces on filaments that arise from a combination of molecular motor forces and interparticle interactions with other contents of the vesicle. In this regard, we consider that momentum conservation in the center-of-mass frame does not play a role in vesicle shape transformations. For example, common models of microorganisms assume a stroke-averaged force dipole model[82,83] or a squirmer model[84,85]. Moreover, to capture the details of individual trajectories at walls, hydrodynamic interactions are important[86,87]. However, based on previous works[88,89], we assume that organization at boundaries with high filament densities can be robustly described by self-propelled particle models. Thus, although our polar active force leads to center-of-mass motion of the vesicle in some states, we do not analyze this behavior in this work.

The vesicle is constructed as a triangulated mesh of $N_{ves} = 2432$ monomers. It has a nominal radius of $R_{ves} \approx 25\sigma$, measured as the distance from the center of mass of the vesicle to the center of any given vesicle monomer in the undeformed state. The diameter of the vesicle monomers is set to $\sigma_{ves} = a\sigma$ with $a \approx 1.934$. As with the filament monomers, the vesicle monomers are bonded through a stiff expanded FENE potential with coefficient $k_{ves}$, with the equilibrium bond length determined by whether the bond is part of a pentameric or hexameric bond. In combination with the increased size of the vesicle monomers, this ensures that active filaments are unable to escape the vesicle. Vesicle curvature is penalized through a harmonic dihedral potential acting on neighboring triangles of the vesicle mesh.

The filament volume fraction in the undeformed state is given by $\phi = N_{fil}V_{fil}/V_{ves}$, where $V_{fil} = \pi\sigma^3/6 + (M-1)\pi b\sigma^2/4$ is the approximate volume of a single filament—accounting for the overlap of bonded monomers—and $V_{ves} = 4\pi R_{ves}^3/3$ is the nominal volume of the vesicle. Note that the volume fraction is defined with respect to the nominal volume of the undeformed vesicle, without considering the finite size of the filament and vesicle monomers. Additionally, since the mesh topology is conserved in our simulations, we model an elastic vesicle; in subsequent work we plan to consider the effects of fluidizing the vesicle and imposing area or volume constraints.

We simulate the coupled Langevin equations for the filament and vesicle bead dynamics using LAMMPS[90], modified to include the active force. We neglect long-ranged hydrodynamic interactions for this system of high filament density; we will investigate their effect in a future study. We present simulation parameters with units such that the mass of all beads is $m = 1$, and energies, lengths, and time are respectively in units of $k_BT$, $\sigma$, and $\tau = \sqrt{m\sigma^2/\epsilon}$. The friction constant is set to $\gamma = 1/\tau$. For additional model details, see the Supplementary Method 1 (See Supplemental Material for model details and additional figures).

## Data availability

The post-processed simulation data generated in this study have been deposited in the Open Science Framework database[91], and is available at osf.io/7s9jp. The raw simulation trajectories data are available from the corresponding authors upon request.

## Code availability

The modified LAMMPs source code used in this study is available through the Open Science Framework database[91], and is available at osf.io/7s9jp.

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

## Acknowledgements

We acknowledge support from NSF DMR-1855914 (MSEP, MFH) and the Brandeis Center for Bioinspired Soft Materials, an NSF MRSEC (DMR-2011846) (MSEP, AB, MFH). We also acknowledge computational support from NSF XSEDE computing resources allocation TG-MCB090163 (Stampede and Comet) (MSEP, MFH) and the Brandeis HPCC which is partially supported by DMR-MRSEC 2011486 (MSEP, AB, MFH) and OAC-1920147 (MFH). We also acknowledge the KITP Active20 program, during which some of these ideas were developed, which is supported in part by the National Science Foundation under Grant No. NSF PHY-1748958 (AB, MFH).

## Author contributions

M.S.E.P., A.B., and M.F.H. designed the research; MSEP performed the computational modeling; M.S.E.P., A.B., and M.F.H. performed the theoretical modeling; M.S.E.P. analyzed the data; and M.S.E.P., A.B., and M.F.H. wrote the paper.

## Competing interests

The authors declare no competing interests.
