## [Peer Review File · Nature Communications]

REVIEWER COMMENTS

Reviewer #1 (Remarks to the Author):

In the manuscript entitled "Vesicle shape transformations driven by confined active filaments" the authors study how the shape of an elastic vesicle changes under the forces of contained self-driven filaments. The work seems to be carefully done, with apparently all the necessary attention to details - like using overlapping beads to avoid a corrugated potential. The manuscript is excellently written, with a well ordered systematic explanation of observed phenomena. The subject of self propelled particles in general, and 'artificial cytoskeleton' are current hot topics of broad interest to physicists and biologists alike. To my knowledge, the presented results are new and exciting. The manuscript thus certainly deserves publication in nature communications.

I have just a few minor comments:

- The authors should acknowledge more clearly the fact of violated momentum conservation. While for 2D Systems, the driving force can be generated against the substrate, the 3D Case (especially enclosed in a vesicle) is a bit more difficult. One might expect eg. Vesicles in the polar-prolate state to display net motion. My argument would be that the propulsion force creates a net reaction force. This reaction force gets transmitted via the other contents of the vesicle, to the vesicle membrane, in which case the vesicle would not move. Alternatively, if the filaments are propelled by external forces the vesicle would be propelled.
- You write " $V_{ves}=4 \pi R^3$ " Dont the vesicle beads also have excluded volume? Thus " $V_{ves}=4 \pi (R-\sigma/2)^3$ " ?
- Please indicate $f_a=8$ also in the Caption of Figure 1
- While I understand your usage of LJ-Units, I prefer to have them explicitly stated in all numbers. Eg. $R_{ves}=25\sigma$. In particular I think providing activity in f_a is a bit misleading. In the field People seem to like the Peclet number and Flexure Number as an indicator of activity. Please state them.
- please provide a more accurate equation for the Persistence length of the path (page 3, top left)
- The manuscript (almost) limits its study to the case of "strong confinement", or large Peclet numbers. This limitation should be made clear. E.g. I expect that the "independence of f_a " found in fig3, would go away if even smaller f_a where studied.
- Placement of Citation [64] suggests an explanation for $\tau \propto L/v_0$, which it is not. Maybe place together with [62,63]
- at some points (eg p4, bottom) you use γ and λ as symbols for surface tension. Because γ is also used for friction, I suggest to consistently stick to λ .
- I did not find any data availability statement. I strongly encourage the authors to publish their data in an open access repository.
- Ref [37] indicates that the membrane friction vs monomer friction plays an important role. Can the authors comment?

Reviewer #2 (Remarks to the Author):

Referee Report: NCOMMS-21-14818

Paper title: Vesicle shape transformations driven by confined active filaments

Paper authors: Peterson, Baskaran, and Hagan

Summary: The submitted manuscript considers active polar filaments confined within a vesicle. The authors employ well-established Langevin dynamics to study this interesting system by exploring the conformation of the containing vesicle and internal filaments. They begin by varying filament aspect ratio and volume fraction, which introduces the five principle vesicle conformations. The article then explores the mechanisms leading to self-organization, with a particular focus on cap morphology. The submission concludes by considering the effect of each flexibility, independently.

Originality: The study is quite original, in my opinion. I appreciated the authors' perspective of activity competing against deformable confining boundaries. I found this study to be an admirable and well-defined approach to explore an interesting bigger question.

Mathematical correctness: The simulation approach is well-established and appears to be implemented well. The models in the supplementary materials appear justified and correct.

Publishability: This manuscript should absolutely be published in the peer-reviewed literature. It is a valuable and well-done computational study. However, I hesitate to say that it is as far-reaching and broadly impactful as I would expect of a manuscript published in Nature Communications. In my opinion, this would certainly be a distinctly excellent article Soft Matter or PRE and it may warrant publication in Nature Communications but I would like to see the authors make a stronger arguments for this in the main text of the article itself.

Recommendation:

1. Revisions, if the authors can make a more compelling argument for publication in a general-readership journal.

Major Comments:

1. As stated above, I think that it is incumbent on the authors to better argue that their study is of broad interest to the readers of Nature Communications. This is my principle concern — I enjoyed the reading the manuscript.

2. I was not convinced by paragraph three of the introduction, any more than I would say the same of any system. Yes, internal effects and boundary conditions determine the behaviour of systems — essentially all systems, not just active systems. That being said, I do certainly agree with the authors that there is something interesting in active systems (which often have self-organized scales) and their confinements (which introduce competitive scales). However, I wasn't convinced by the argument presented here. I think this might need to be re-posed. In someways, I think this connects to my comment on the publishability: The science is strong but placing it in the most impactful context seems essential if it is to be accepted here.

3. In Methods, the simulations of the filaments is described in detail but the vesicle is not described at all. I see that details are given in the SI but vesicle is on equal footing to the filaments in this study. What is the mesh? Is vesicle rigidity the same as the filaments? Is it implemented in the same way? Why can't filaments go through the pores in the mesh?

4. I don't feel like I know anything about the dynamics of this system. How long lived are the states and do they have a probability of switching between states? For instance, how likely is II III or III IV ? Is II IV possible or must the system pass through state III? These seem like important questions that are thoroughly ignored.

0. I understand that state-III translates and IV does not — but does IV spin?

1. In Figure 3 (and the discussion of it), there is a threshold active force $f_a \lesssim 1$. However, the presented data seems to be only for $f_a \in \{0, 1, 2\}$, with the $f_a = 1$ mostly exhibiting the strong force behaviour (or at least non-spherical vesicles). Although it is certainly not the most exciting transition that the authors consider, it is in many ways the baseline and I'd like to see this nailed down better with a finer step size between 0 and 1.

2. While reading your discussion of the competition between time scales, I kept wanting to think about the ratio as a Péclet number, but I realize that's not quite right. The time τ_{rot} isn't a rotational diffusion but is a propulsive timescale, ("timescale equal to the that over which the rod would have moved its own length"). On first reading of the manuscript, the rotational diffusion time scale appears to be ignored; however, on second reading it is including in the guise of persistence length and in the supplementary discussion as τ_{corr} . I had to work through the aspect ratio for filaments in Figure 3 and compare it to $a_{SC} = 4.3$ in the SI to be sure that it was negligible. I think the authors should include this third timescale in the discussion of τ_{rot} and τ_{coll} .

3. It is not clear to me how a^* is found. It is shown in Figure 4 but the text states that $a^* = 130$ is found from the data in Figure 1, which presumably is only a subset of the data shown in Figure 4 (the small value limit, perhaps?). This is currently unclear.

Minor Comments:

1. I'm not sure whether or not it would be worth including in the manuscript, but I would be interested to see Figure 4 on a log-log graph.

2. The first sentence of the abstract reads "... deformable boundaries provide a mechanism to organize internal active stresses and perform work on the external environment." Is the intent of this sentence that deformable boundaries act as the mechanism by which work is done on the external environment? I don't find this very clear and, besides, the manuscript isn't about the vesicles performing work on the external environment. I would like to see this sentence re-worked.

3. In the introduction, the second half of a sentence reads "confining active particles leads to system-spanning effects" but the issue is that the beginning of the sentence cites "bands or flocks" as examples of phenomena driven by anisotropic interactions. But bands are system-spanning, in at least some sense. I feel this weakens the distinction the authors are trying to make.

4. In the same sentence, it is written "... changing the length and stiffness of active polymers leads to dramatic reorganization of active stresses..." It seems to me that this point could be made stronger by citing the work of the Gompper group that considers active filaments. Similarly, I was surprised that the role of Frank elasticity in active nematics wasn't used to strengthen this argument.

5. Similarly, I would have liked to see the authors compare and differentiate their work from that of studies of active nematic droplets. It's clear that they aren't the same (vesicle stiffness is not surface tension; self-propelled rods are not active nematic fluids); however, discussing the extent of these similarities and differences might be helpful.

6. It is written "... simulations on infinitely rigid vesicles did not exhibit..." These can't really be infinitely rigid. Can this please be said more precisely?

7. The Figure 1 caption should state $f_a = 8$.

Reviewer #3 (Remarks to the Author):

The authors report on the non-equilibrium shapes, and dynamics, of vesicles that enclose active semi-flexible rod. The resulting self-organization of the rods, and the subsequent deformation of the membrane, is interesting. Its an extension to 3D of previous works of very similar nature that were performed in 2D. The change in dimensionality makes this study more relevant to understanding the shapes of biological cells, and artificial vesicles. In its current form I find the work interesting but incomplete in its present form.

Comments:

1) They miss a very relevant reference on experiments with actin filaments that deform vesicles, and where the funneling effect of the membrane on the filaments was observed:

Liu, A. P., Richmond, D. L., Maibaum, L., Pronk, S., Geissler, P. L., & Fletcher, D. A. (2008).

Membrane-induced bundling of actin filaments. *Nature physics*, 4(10), 789-793.

2) The method for computing the vesicle shape is not given at all in the main text, it only says "elastic vesicle". I was under the impression it has only bending energy, but then in Fig.6 i find that it has a Young's modulus. This is not clear.

3) The bending modulus used in Fig.1 is not given in the caption, and similarly there is missing information in many later figures.

4) On page 3 we are sent to Fig.6b: What is the source of the feedback between the active filaments and vesicle flexibility that organizes the rotating ring ? Some suggestions and a deeper investigation of this feedback is needed. Is it that the furrow formed by the band "protects" it from running into bands moving in the opposite direction ?

5) The appearance of caps of finite size, which are dynamic, is reminiscent to the spontaneous size of the clusters of active proteins at the tips of protrusions, as shown in:

Fošnarič, Miha, et al. "Theoretical study of vesicle shapes driven by coupling curved proteins and active cytoskeletal forces." *Soft Matter* 15.26 (2019): 5319-5330.

6) The vesicles in the present study seem not to allow for large deformations, such as narrow protrusions. It seems therefore that there are different classes of shapes that they did not explore here, either using softer vesicles, or larger forces.

7) The snap-shots in Fig.4: please explain which volume fraction or L_{rod} values have been changed to go from one to the other.

8) In Fig.5b it seems that large f_a inflates the vesicle, so could this increase in volume reduce the volume fraction enough to cause a transition from caps to disordered/uniform phase ?

9) In Fig.5b it seems that increase in f_a decreases the number of caps, and this is not described by eq.2. Can this be rationalized ?

10) The linear dependence on the 3D volume fraction ϕ in Eq.2 seems strange: intuitively the number of caps should depend on the areal density of rods on the vesicle membrane, so should depend on $\phi^{2/3}$? it will also fit much better the simulation data.

11) There is no dependence in Eq.2 on the bending modulus of the vesicle. How does it affect the number of caps ? See also the dependence of the size of the protrusions on the bending modulus in the reference given in point 5.

12) The bending modulus of the membrane should be written in Fig.6b.

13) What is the Young's modulus of the vesicle ? Why isnt the threshold of the FvK denoted on Fig.6a ?

14) In Fig.6a there are ordered caps in faceted vesicles, (volume fraction 0.05, low k_{ves}) which seem to change to a ring at high κ . Why ? this is not clear.

Why arn't the filaments aggregating in the sharp corners of faceted vesicles (Fig.6a low volume fractions) ?

15) Missing relevant ref:

Nikola, N., Solon, A. P., Kafri, Y., Kardar, M., Tailleur, J., & Voituriez, R. (2016). Active particles with soft and curved walls: Equation of state, ratchets, and instabilities. *Physical review letters*, 117(9), 098001.

We thank the reviewers for their careful reading of the manuscript and constructive critiques. We have significantly revised the manuscript in response to these comments, and we believe that it now describes our results much more clearly and comprehensively. Especially important critiques were: Reviewer 1 asked us to clarify the description of our simulated dynamics in the main text and the behavior outside of the strong confinement limit, Reviewer 2 felt that the manuscript did not sufficiently explain the broad appeal of our study and in particular what is special about active systems in our context, and Reviewer 3 asked for a more in-depth discussion on the difference between our work and previous studies.

In response to these comments we have extensively rewritten the introduction to (1) more clearly explain the dynamics, its limitations, and the motivation for using a minimal model; (2) to explain the fundamental aspects of our study, including why the effects of internal stresses and boundary conditions are fundamentally different in an active material compared to equilibrium; and (3) we have included references that we overlooked in the previous version as well as a more detailed explanation of the important ways in which our study builds on that previous work. We have also added additional discussion in the results section comparing our findings to those of previous work. Further, we have added discussion and updated figures (Fig. 3 and Fig. S2) explaining the low-activity behavior when the system falls outside of the strong confinement limit, and information about the dynamics and timescales associated with reaching steady state (new Fig. S4).

We present detailed responses to all critiques along with our revisions to the manuscript point-by-point below, and we have included a version of the manuscript with tracked changes in the upload.

Reviewer #1 Comments

- 1. The authors should acknowledge more clearly the fact of violated momentum conservation. While for 2D Systems, the driving force can be generated against the substrate, the 3D Case (especially enclosed in a vesicle) is a bit more difficult. One might expect eg. Vesicles in the polar-prolate state to display net motion. My argument would be that the propulsion force creates a net reaction force. This reaction force gets transmitted via the other contents of the vesicle, to the vesicle membrane, in which case the vesicle would not move. Alternatively, if the filaments are propelled by external forces the vesicle would be propelled.*

We agree. Although our vesicles do move in the lab frame in many of the steady states, our study focuses on stress organization at a deformable boundary through contact interactions. For this purpose, the momentum conservation in the center of mass frame is irrelevant. For example, when considering microorganisms, it is typical to consider a stroke-averaged force dipole model (i.e., Refs. 57 and 58) or a squirmer model (i.e., Refs. 59 and 60). To capture the details of individual trajectories at walls, the hydrodynamics is important (i.e., Refs. 61 and 62). But when one seeks to understand organization at boundaries at high densities, the relevant phenomenology is robustly captured by self-propelled particle models (i.e., Refs. 63 and 64). It is in this spirit that we study this very simple model. Moreover, we have intentionally not

focused on center of mass dynamics in our results, as we agree with the reviewer's point that these are unphysical and unreliable without proper momentum conservation and capturing the physics of a 3D wet system.

Modification: We have removed references to the vesicle motion in the results section of the manuscript, and now make this feature/limitation of this work more explicit in the revised manuscript in the second paragraph of the Methods section.

2. You write " $V_{ves}=4\pi R^3$ " Don't the vesicle beads also have excluded volume? Thus " $V_{ves}=4\pi(R-\sigma/2)^3$ " ?

Indeed, both the vesicle and filament monomers have excluded volume. However, this is only a nominal volume; since the vesicle does not maintain a perfectly spherical shape, it might not be useful to be this precise in the available volume for the filaments. Additionally, this only changes the volume by an order $O(R^2)$ correction, which is not a significant difference.

Modification: We have added a sentence to the Methods section clarifying this point:

Note that the volume fraction is defined with respect to the nominal volume of the undeformed vesicle, without considering the finite size of the filament and vesicle monomers.

3. Please indicate $fa=8$ also in the Caption of Figure 1.

Modification: We have added the value of fa to the Fig. 1 caption.

4. While I understand your usage of LJ-Units, I prefer to have them explicitly stated in all numbers. Eg. $R_{ves}=25\sigma$. In particular I think providing activity in fa is a bit misleading. In the field People seem to like the Peclet number and Flexure Number as an indicator of activity. Please state them.

Modification: We have switched to using explicit values with units for consistency, as well as defining the control parameters for activity (Peclet number, Pe) and filament/vesicle stiffness (rigidity, χ).

While we agree that the flexure number is a useful parameter, we do not think that it is the natural control parameter for our system. In particular, as we note in the Supplemental Material (Sec. E), we find that the filament persistence length is renormalized by activity so that it scales as $1/Pe^2$, and therefore is not adequately captured by the flexure number. Instead, we define a rigidity parameter χ that scales with the passive filament persistence length, which leads to a more intuitive expression for the critical value of Pe below which the filaments are no longer in the strong confinement limit.

5. Please provide a more accurate equation for the Persistence length of the path (page 3, top left).

Modification: We have fixed the form of the equation for the persistence length of the path both in the main text and in the Supplemental Material.

6. *The manuscript (almost) limits its study to the case of "strong confinement", or large Peclet numbers. This limitation should be made clear. E.g. I expect that the "independence of f_a " found in fig3, would go away if even smaller f_a were studied.*

We agree that at small values of f_a we should see a dependence on the activity, this was stated in the main text of the original manuscript.

Modification: In response to this comment, we have performed additional simulations and analysis at low values of the activity parameter and included this data in Fig. 3. As expected, there is a weak dependence on activity at low values of f_a . We also show an additional snapshot in Fig. S2.

7. *Placement of Citation [64] suggests an explanation for tau propto L/v_0 , which it is not. Maybe place together with [62,63].*

Thank you for pointing out this misplaced footnote.

Modification: We've moved [64] (now [77]) to a more relevant location.

8. *At some points (eg p4, bottom) you use gamma and lambda as symbols for surface tension. Because gamma is also used for friction, I suggest to consistently stick to lambda.*

Modification: We have corrected the inconsistent usage of lambda for the interfacial tension parameter.

9. *I did not find any data availability statement. I strongly encourage the authors to publish their data in an open access repository.*

Modification: We have added data availability and code availability statements. We have posted the modified LAMMPS source code, all of the analysis scripts, and the data files used in the study on the Open Science Framework site. Because of the large size of the raw simulation trajectories (2 TB) we are making it available upon request rather than directly posting it on the Open Science Framework Site.

10. *Ref [37] indicates that the membrane friction vs monomer friction plays an important role. Can the authors comment?*

In Ref. [37] (now Ref. [45]) they do not study or obtain conclusions about the effects of friction between membrane and filament monomers; rather, they find that friction between the membrane and the substrate for interface that the flexocyte is crawling on plays an important role. Varying membrane-filament friction could indeed be interesting, but given the already very large parameter space that characterizes our system, it is beyond the scope of the present article.

Reviewer #2 Comments

Major Comments

1. *As stated above, I think that it is incumbent on the authors to better argue that their study is of broad interest to the readers of Nature Communications. This is my principle concern — I enjoyed the reading the manuscript.*
2. *I was not convinced by paragraph three of the introduction, any more than I would say the same of any system. Yes, internal effects and boundary conditions determine the behaviour of systems — essentially all systems, not just active systems. That being said, I do certainly agree with the authors that there is something interesting in active systems (which often have self-organized scales) and their confinements (which introduce competitive scales). However, I wasn't convinced by the argument presented here. I think this might need to be re-posed. In some ways, I think this connects to my comment on the publishability: The science is strong but placing it in the most impactful context seems essential if it is to be accepted here.*

Thank you for highlighting this. We agree that the original manuscript did not sufficiently explain what is special about active systems in our context, and in particular why a model that accounts for internal stress organization, boundary forces, and feedback between the two is of broad interest. We will break our response to this critique and the corresponding changes to the manuscript into two parts.

A. Why active systems are special. In the original version of the manuscript, we argued that there are two mechanisms for organizing active stress, inter-particle interactions that realign active forces and deformable boundaries that exert passive reaction stresses that further shape and organize active stress. As the reviewer points out, all systems experience effects due to internal stresses and boundaries. However, as also pointed out by the reviewer, in contrast to the systems traditionally studied in materials physics, active systems lack a separation of scales between correlations and observables. Consequently, the effects of boundaries are non-extensive, and can qualitatively change the macroscale behaviors of an active system. Moreover, the processes that lead to macroscale observables are statistical at every relevant scale in an active material. Energy is continuously input at the component-scale, which drives active forces that self-organize into mesoscale active stresses in a manner that depends on the local physical environment; in turn, these active stresses drive macroscale behaviors. Thus, the relationship between organization of internal stresses and macroscale observables fundamentally differs in active systems, in comparison to traditional materials.

Previous works that studied these effects independently have already established unique aspects of active materials. However, currently we know little about what happens when both mechanisms occur simultaneously and can couple with each other. Our study addresses this in the context of a minimal microscopic model.

Modification: Based on the reviewer's critique, we have now modified the introduction to more clearly explain (while trying to maintain brevity) why the behaviors of internal stress organization and boundaries are fundamentally different in active systems, and why it is important to learn what happens when these effects couple to each other.

B. Even though active systems are special, why should someone outside of this immediate field care about coupling between internal stress organization and boundary forces? The case we make is the following: dynamical organization of active stress is the physical mechanism at play in diverse cellular processes, from cell division to endocytosis and motility. Thus, understanding this phenomenon is of fundamental interest both to cell biology and nonequilibrium statistical mechanics. Moreover, learning to harness and design such capabilities in biomimetic or synthetic systems would lead to transformative new applications, making this study of interest to disciplines across physics, chemistry, and materials science. However, identifying and understanding transferable design principles for the organization of active stresses is not achievable in the context of complex and faithful models of cytoskeletal dynamics. Thus, minimal mechanical models are essential for this exploration. Further, the complexity of the emergent behavior in this intrinsically nonequilibrium class of materials warrants developing models that allow us to ask specific questions. In our case, the question is: what are the physical mechanisms underlying shape transformations in vesicles deformed by active stresses?

Modification: We have extensively modified the introduction to hopefully better emphasize these points.

3. *In Methods, the simulations of the filaments are described in detail, but the vesicle is not described at all. I see that details are given in the SI but vesicle is on equal footing to the filaments in this study. What is the mesh? Is vesicle rigidity the same as the filaments? Is it implemented in the same way? Why can't filaments go through the pores in the mesh?*

We agree that the description of the vesicle was underdeveloped in the main text.

Modification: We have added the following paragraph to the Methods section that discusses the vesicle in much more detail:

The vesicle is constructed as a triangulated mesh of $N_{ves} = 2432$ monomers. It has a nominal radius of $R_{ves} \approx 25\sigma$, measured as the distance from the center of mass of the vesicle to the center of any given vesicle monomer in the undeformed state. The diameter of the vesicle monomers is set to $\sigma_{ves} = a\sigma$ with $a \approx 1.934$. As with the filament monomers, the vesicle monomers are bonded through a stiff expanded FENE potential with coefficient k_{ves} , with the equilibrium bond length determined by whether the bond is part of a pentameric or hexameric bond. In combination with the increased size of the vesicle monomers, this ensures that active filaments are unable to escape the vesicle. Vesicle curvature is penalized through a harmonic dihedral potential acting on neighboring triangles of the vesicle mesh.

4. *I don't feel like I know anything about the dynamics of this system. How long lived are the states and do they have a probability of switching between states? For instance, how likely is $II \rightleftharpoons III$ or $III \rightleftharpoons IV$? Is $II \rightleftharpoons IV$ possible or must the system pass through state III? These seem like important questions that are thoroughly ignored.*

We agree that the dynamics of this system is an important aspect for understanding of the interplay between self-organizing active agents and deformable boundaries. However, given the large parameters set and the complexity of behaviors already present at steady state, we feel that analyzing the transient approach to steady-state falls beyond the scope of the current paper. Therefore, we have limited our focus on understanding the long-lived dynamical steady-states which will help set the foundation for future studies of this system that look at the dynamics of these states.

In general, we find that the dynamical steady states are very long-lived compared to the total simulation time, and reconfiguration events are quite rare.

Modification: We have added an additional figure to the Supplemental Material (Fig. S4) that better shows this. We have additionally added the following text to SI Sec. F:

The first 10% of the simulation is used for initialization, during which the filament positions and orientations are randomized. The final 10% of the simulation is used to analyze the resulting vesicle conformations and filament organizations. Note that the system reaches a long-lived steady state configuration very rapidly, and tends to exist in that state for nearly the entirety of the simulations (see Fig. S4).

5. *I understand that state-III translates and IV does not — but does IV spin?*

Yes, we find that all cap-forming states can spin. However, as pointed out by Reviewer 1, our system does not conserve momentum or account for hydrodynamic interactions. As a result, we have decided to de-emphasize the net vesicle motion as it may be confusing to readers.

Modification: We have added the following text in the Results section:

We note that states with net polarity can exhibit center-of-mass motion, but more comprehensive models that account for momentum conservation would be important to study such effects. Therefore, in what follows we focus on the internal motions and shape transformations of the vesicle.

6. *In Figure 3 (and the discussion of it), there is a threshold active force $f_a \lesssim 1$. However, the presented data seems to be only for $f_a \in \{0, 1, 2\}$, with the $f_a = 1$ mostly exhibiting the strong force behaviour (or at least non-spherical vesicles). Although it is certainly not the most exciting transition that the authors consider, it is in many ways the baseline and I'd like to see this nailed down better with a finer step size between 0 and 1.*

This is a good point (Please also see our response to Rev. 1 comment 6).

Modification: We have performed additional simulations (shown in Fig. 3) that better showcase the weak dependence on activity for small values of f_a . We have also placed an additional snapshot in Fig. S2 to make the “ragged caps” more apparent.

7. *While reading your discussion of the competition between time scales, I kept wanting to think about the ratio as a Peclet number, but I realize that's not quite right. The time τ_{rot} isn't a rotational diffusion but is a propulsive timescale, ("timescale equal to the that over which the rod would have moved its own length"). On first reading of the manuscript, the rotational diffusion time scale appears to be ignored; however, on second reading it is including in the guise of persistence length and in the supplementary discussion as τ_{corr} . I had to work through the aspect ratio for filaments in Figure 3 and compare it to $a_{SC} = 4.3$ in the SI to be sure that it was negligible. I think the authors should include this third timescale in the discussion of τ_{rot} and τ_{coll} .*

We see that a lack of explicit mention of this timescale in the text was confusing in the previous version. Based on the observation that cap and ring formation does not occur outside of the strong confinement limit, and to simplify the presentation, we have assumed that the system is in the strong confinement limit during this discussion. This implies that the rates of reorientation and collision are very fast compared to the rate of rotational diffusion. While the assumption of the strong confinement limit was mentioned in the previous version, we agree that we should make its implication on the rotational timescale clearer.

Modification: We have added the following text just before the timescale analysis:

In the following discussion we assume the strong confinement limit; in particular, we assume that the timescale governing the rotational diffusion of the filaments, τ_{coll} , is long compared to the other relevant timescales.

Modification: For additional clarity, we also add the following sentence after our timescale analysis:

That is, we assume that the rotational diffusion timescale of the filaments, τ_{corr} , is much larger than both τ_{rot} and τ_{coll} .

8. *It is not clear to me how a^* is found. It is shown in Figure 4 but the text states that $a^* = 130$ is found from the data in Figure 1, which presumably is only a subset of the data shown in Figure 4 (the small value limit, perhaps?). This is currently unclear.*

Modification: We have now clarified in the caption how we estimated a^* .

Specifically, the data in Fig. 4 consists of all data points from Fig. 1 in which the system forms rings or caps. To estimate a^* , we fit a line to the data points with $n_{cap} \leq 7$ (we did not include data points with more caps because the identification and counting becomes unreliable).

9. *I'm not sure whether or not it would be worth including in the manuscript, but I would be interested to see Figure 4 on a log-log graph.*

We show here Fig. 4 on a log-log scale (removing the points corresponding to rings which have no caps, as they are not representable on a log scale). We have decided not to include this in

the manuscript, as we think it doesn't show the trend any better than that linear scale plot, and it does not allow us to include the ring state.

10. *The first sentence of the abstract reads “. . . deformable boundaries provide a mechanism to organize internal active stresses and perform work on the external environment.” Is the intent of this sentence that deformable boundaries act as the mechanism by which work is done on the external environment? I don't find this very clear and, besides, the manuscript isn't about the vesicles performing work on the external environment. I would like to see this sentence re-worked.*

We agree that the intent of this sentence was unclear. Our original thought was that, in biological contexts, the active material is often contained within a deformable container, and thus the container acts as the mediator of interactions between active material and the external environment; however, as pointed out by the reviewer, this is tangential to the focus of this paper.

Modification: We have removed the portion of this sentence that mentions performing work so that it more clearly emphasizes the role of the deformable boundary on the organization of active stresses, which is the main focus of this work.

11. *In the introduction, the second half of a sentence reads “confining active particles leads to system-spanning effects” but the issue is that the beginning of the sentence cites “bands or flocks” as examples of phenomena driven by anisotropic interactions. But bands are system-spanning, in at least some sense. I feel this weakens the distinction the authors are trying to make.*

The point of this sentence is not to say that confinement and anisotropic interactions are mutually exclusive in their ability to create system-spanning effects. In fact, it is exactly because both can serve as mechanisms of generating large-scale behaviors that we are interested in combining them in this work.

Modification: We have modified the text in this paragraph to clarify our meaning.

12. *In the same sentence, it is written “. . . changing the length and stiffness of active polymers leads to dramatic reorganization of active stresses. . . ” It seems to me that this point could be made stronger by citing the work of the Gompper group that considers active filaments. Similarly, I was surprised that the role of Frank elasticity in active nematics wasn’t used to strengthen this argument.*

Modification: We have added additional references from Gompper and co-workers here.

While we agree that Frank elasticity is very relevant in this context for bulk simulations of active filaments, it’s not clear if there are well-defined Frank elastic constants for our system, due to the fact that it is dilute and undergoes a number of structural transformations. Hence, we have not explicitly mentioned it in this work in order to avoid confusion.

13. *Similarly, I would have liked to see the authors compare and differentiate their work from that of studies of active nematic droplets. It’s clear that they aren’t the same (vesicle stiffness is not surface tension; self-propelled rods are not active nematic fluids); however, discussing the extent of these similarities and differences might be helpful.*

This is a good point, we did not do a good job of emphasizing the novel features of our simulations in comparison to the previous studies on active droplets. We are only aware of a few studies of active droplets, which are now cited as Refs. 47 – 50. These works provide great motivation for our study, but as pointed out by the reviewer the physics is quite different. Most importantly, the theoretical formulations used in these works necessarily require assumptions about the types of organization the active particles can undergo as well as the form of the hydrodynamic boundary conditions. In contrast, the filament organization and interaction with the membrane is an emergent property in our computational model, which as we show leads to filament organizations that could not be considered in these previous works.

Modification: We now summarize these differences in the introduction:

More closely related to our work are simulation studies of droplets containing active material that show tantalizingly life-like behaviors such as motility and division [47–50]. These elegant studies highlight the importance of understanding the types of emergent behaviors that arise when active matter and deformable boundaries are combined. However, the continuum hydrodynamic theories employed in these works require key assumptions about the nature of particle organization and particle-membrane interactions.

14. It is written “. . .simulations on infinitely rigid vesicles did not exhibit. . .” These can't really be infinitely rigid. Can this please be said more precisely?

We did not make this point sufficiently clear in the original version of the manuscript. Our infinitely rigid vesicle is indeed infinitely rigid; we implement it by fixing the positions of the vesicle monomers in their initial spherical configuration so that the active filaments are unable to deform the vesicle.

Modification: We have added a sentence in the Supplemental Material (Sec. F) that clarifies this:

Infinitely rigid vesicles are implemented by not integrating the equations of motion for the vesicle monomers so that they are always in their initial (spherical) configuration.

15. The Figure 1 caption should state $f_a = 8$. (See also Reviewer #1 Comment 3)

Modification: We have added the value of f_a to the Fig. 1 caption.

Reviewer #3 Comments

1. *They miss a very relevant reference on experiments with actin filaments that deform vesicles, and where the funneling effect of the membrane on the filaments was observed:
Liu, A. P., Richmond, D. L., Maibaum, L., Pronk, S., Geissler, P. L., & Fletcher, D. A. (2008).
Membrane-induced bundling of actin filaments. Nature physics, 4(10), 789-793.*

Thank you for bringing this very relevant work to our attention.

Modification: We have included it in our manuscript as Ref. 12.

2. *The method for computing the vesicle shape is not given at all in the main text, it only says "elastic vesicle". I was under the impression it has only bending energy, but then in Fig.6 I find that it has a Young's modulus. This is not clear.*

Modification: We have added additional information about the vesicle model to the main text to make this clearer.

Please also see Reviewer 2 Comment 3 for further details.

3. *The bending modulus used in Fig. 1 is not given in the caption, and similarly there is missing information in many later figures.*

Modification: We have added additional information about the various system parameters used in each state-diagram figure.

4. *On page 3 we are sent to Fig.6b: What is the source of the feedback between the active filaments and vesicle flexibility that organizes the rotating ring? Some suggestions and a deeper investigation of this feedback is needed. Is it that the furrow formed by the band "protects" it from running into bands moving in the opposite direction?*

From this comment we see that our description of the polar rings was not sufficiently clear. In this state, the filaments form a single band, with all filaments moving in the same direction. In a rigid vesicle system, such a ring is unstable to fluctuations of the filaments' orientations in a direction orthogonal to their direction of motion, as is shown in Fig. 6b (as is expected from linear stability analysis). However, a flexible vesicle is able to provide a restoring force that stabilizes these transverse fluctuations, allowing the polar ring to exist as a stable dynamical steady-state.

Modification: We have added additional clarifying text in our description of the oblate spheroid states (II) within the Results section:

*[...] This transition is driven by the filaments organizing into a stable polar band in **which all filaments move in the same direction, resulting in deformations** of the vesicle along a geodesic. [...] However, note that such polar bands would be **unstable to fluctuations that are transverse to the polar orientation of the band***

*[...] which in turn provides a restoring force to stabilize **transverse fluctuations** to filament alignment within the band.*

5. *The appearance of caps of finite size, which are dynamic, is reminiscent to the spontaneous size of the clusters of active proteins at the tips of protrusions, as shown in: Fošnarič, Miha, et al. "Theoretical study of vesicle shapes driven by coupling curved proteins and active cytoskeletal forces." *Soft Matter* 15.26 (2019): 5319-5330.*

Thank you for bringing this interesting article to our attention. We agree that their results motivate the importance of deformable boundaries on organizing active stresses, and that the budding in that work is qualitatively reminiscent to the capping behavior that we see in our system. However, the physics underlying these behaviors is quite different. In their work they are using an active protein model with attractive, isotropic inter-protein interactions, in contrast to our filament model with purely repulsive, anisotropic interactions. Thus, the protein aggregation they see on their membrane results from explicit energetic attractions, whereas our filament aggregation is an activity-induced effect. Moreover, in their work the proteins are constrained to the membrane surface and the protein orientations are constrained always to be orthogonal to the membrane. In our simulations we do not constrain the filament positions or orientations, and thus the active stresses can be applied to the vesicle both in the direction of the vesicle normal as well as in the tangent plane. Hence, the cap formation is entirely an activity-induced emergent property in our system.

Modification: This work provides excellent motivation for our work, and we have added it as Ref. 35. We also summarize the similarities and differences between in the Results section:

Note that the formation of these caps is qualitatively similar to the buds that were observed in simulations of surface-bound proteins, which have attractive interaction-induced aggregation and exert forces in the normal direction of a highly-deformable membrane [35]. In contrast, in our system the filaments interact only through excluded volume and are not constrained to the membrane; thus, caps are an emergent property arising from activity, excluded volume, and passive forces from the membrane.

6. *The vesicles in the present study seem not to allow for large deformations, such as narrow protrusions. It seems therefore that there are different classes of shapes that they did not explore here, either using softer vesicles, or larger forces.*

We agree that there exists a large class of other interesting vesicle shape transformations that we have not explored in the present work. Due to the faceting transition of the vesicle at a critical Foppl-von Karman number, we must extend the vesicle model to allow for fluidization of the vesicle to explore shape changes in highly-deformable vesicles. While this is a potentially fascinating line of exploration (which we are currently working on), we feel that it is beyond the scope of the present manuscript, which already encompasses a very large parameter space.

7. *The snap-shots in Fig. 4: please explain which volume fraction or L_{rod} values have been changed to go from one to the other.*

The snapshots shown in Fig. 4 are example images from simulations used in Fig. 1. Both filament aspect ratio and volume fraction are changed between snapshots.

Modification: We now explain this in the figure caption.

8. *In Fig. 5b it seems that large f_a inflates the vesicle, so could this increase in volume reduce the volume fraction enough to cause a transition from caps to disordered/uniform phase?*

It is correct that very high values of f_a lead to vesicle expansion, with vesicle volume changing by $\sim 100\%$ and area by $\sim 70\%$ in extreme case). While in principle this might provide a mechanism to destabilize caps, we have not observed this, and we think it will not occur for the following reasons. First, this expansion only occurs when all of the filaments are orthogonal to the membrane, i.e., when there are well-formed caps. Second, as we discuss in the response to your next two comments, as long as we are in the strong confinement limit (which we always are at high values of f_a), cap formation depends on the area fraction of filaments (i.e N/A with N the number of filaments and A the vesicle surface area), because all of the filaments are always on the surface. Third, at these high values of f_a the system is far beyond the threshold for cap stability, so much more significant decreases in the area fraction would be required to destabilize the caps.

Modification: As discussed next, we have adjusted the form of Eq. 2 and we think this will now be clearer.

9. *In Fig. 5b it seems that increase in f_a decreases the number of caps, and this is not described by Eq. 2. Can this be rationalized?*

This effect arises because increasing f_a reduces the effective filament bending modulus, leading to an effective decrease in filament aspect ratio. This effect could be explicitly included in this theory by the approach described in Sec. II.C and SI Sec. D for the effects of activity and filaments semi-flexibility on the phase boundary. We have not included this for simplicity and because it does not qualitatively change the conclusions.

Modification: However, we now explicitly note this:

Accounting for this activity-renormalized filament flexibility can also be used in Eq.2 to explain the reduction in the number of caps as activity is increased at a fixed filament rigidity.

10. *The linear dependence on the 3D volume fraction ϕ in Eq. 2 seems strange: intuitively the number of caps should depend on the areal density of rods on the vesicle membrane, so should depend on $\phi^{2/3}$? it will also fit much better the simulation data.*

Since cap formation only occurs well into the strong confinement limit, the dependence is actually on the total number of filaments N and the vesicle surface area, since all filaments are on the surface. Because we are at constant V , this is the same as depending linearly on Φ . We did check a dependence on $\phi^{(2/3)}$, but we find this does not fit the data as well as Eq. 2.

Modification: We have made this relation more apparent by changing the form of Eq. 2 to explicitly refer to the filament number.

11. *There is no dependence in Eq. 2 on the bending modulus of the vesicle. How does it affect the number of caps? See also the dependence of the size of the protrusions on the bending modulus in the reference given in point 5.*

There is no explicit dependence on bending modulus in Eq. 2 because, for simplicity, we have assumed that the local curvature of the vesicle in the vicinity of a cap is approximately equal to the undeformed vesicle curvature $1/R_v$. This assumption holds well enough for the default bending modulus value that we have mainly focused on in this work.

However, it is the case that the local curvature depends on a balance between the active forcing in the vesicle bending modulus, so we expect the local curvature radius to become smaller for smaller values of the bending modulus.

Modification: We now explicitly note that this dependence exists in the main text after Eq. 2:

[...] where $a^ \propto R_v/\sigma$ is an adjustable parameter that depends on the local vesicle curvature at the cap, which results from a balance between active forces from the filaments and passive forces from vesicle bending (see the SI). Thus a^* may depend on the moduli of the vesicle and filaments as well as activity in some limits.*

Modification: We also explicitly mention this dependence in the SI under the heading “Dependence of a^* on activity and bending modulus”.

12. *The bending modulus of the membrane should be written in Fig. 6b.*

Modification: We have added the bending modulus in Fig. 6b.

13. *What is the Young's modulus of the vesicle? Why isn't the threshold of the FvK denoted on Fig. 6A?*

Modification: We have added additional details about the vesicle model to the Methods section of the manuscript (please see response to Reviewer 2 Comment 3 for details). Additionally, we have defined the Young's modulus of the vesicle in terms of the stretching coefficient in the Results section where we discuss Fig. 6a:

[...] it drives a faceting transition when the Föppl–von Kármán number, $FvK = Y R_{ves}^2/k$ where $Y = 2k_{ves}/\sqrt{3}$ [...]

14. In Fig. 6a there are ordered caps in faceted vesicles, (volume fraction 0.05, low k_{ves}) which seem to change to a ring at high kappa. Why? This is not clear. Why aren't the filaments aggregating in the sharp corners of faceted vesicles (Fig. 6a low volume fractions)?

The vesicle facets are still reasonably smooth, enough so that the filaments don't get trapped for very long, and there aren't enough filaments for clusters to form before the filaments are able to free themselves. Thus, caps only form at sufficiently high densities. Our understanding is that the rings form instead as kappa is increased because the facet corners become less sharp, so clusters don't have the natural nucleation points that the facets provide.

Modification: We now explain this in SI Sec. F:

Note that for vesicle rigidities such that faceting occurs ($FvK > \sim 154$ [82]), the facets can act as nucleation points for cap formation, making caps more likely to form rather than, for example, ring states, as is seen in Fig. 6a.

15. Missing relevant ref:

Nikola, N., Solon, A. P., Kafri, Y., Kardar, M., Tailleur, J., & Voituriez, R. (2016). Active particles with soft and curved walls: Equation of state, ratchets, and instabilities. Physical review letters, 117(9), 098001.

Thank you for bringing this work to our attention.

Modification: We have referenced it in the manuscript as Ref. 44.

REVIEWERS' COMMENTS

Reviewer #1 (Remarks to the Author):

The authors have addressed my concerns. The Manuscript is now ready for publication.

Reviewer #2 (Remarks to the Author):

The authors have addressed my comments and I would fully support publication of this exciting manuscript in Nature Communications.

A minor comment: personally, I found log-log scale of Fig. 4 to be very impressive and far more revealing than the linear version presented in the manuscript. But I concede that this is to a certain degree a matter of preference.

Reviewer #3 (Remarks to the Author):

The authors have greatly improved the manuscript, and many questions regarding their observations are now more clearly explained. I recommend publication, but have these additional small comments:

1) Regarding my previous question about the stabilization of the rings along the rim of the oblate-shaped vesicle: can they calculate the balance of forces between the outwards radial push of the rotating filaments and the restoring force of the bent rim of the flattened vesicle ?

This oblate shape is similar in shape and balance of forces to the pancake-like shapes found in [35], using a different self-organization mechanism.

2) In the introduction they should explicitly mention the main example of cellular bio-polymer that affects the membrane shape, ie actin. In this context, they could explicitly mention that [12] found that actin filaments tend to "funnel" into a local deformation, as they find with their caps.

Also, there are several actin-specific simulations that seem relevant enough to mention, here are some that I found:

Ni, Haoran, and Garegin A. Papoian. "Membrane-medyan: Simulating deformable vesicles containing complex cytoskeletal networks." *bioRxiv* (2021).

Zhuravlev, Pavel I., Longhua Hu, and Garegin A. Papoian. "Computer Simulations of Mechano-Chemical Networks Choreographing Actin Dynamics in Cell Motility." *Computational Modeling of Biological Systems*. Springer, Boston, MA, 2012. 231-256.

Huber, Florian, Josef Käs, and Björn Stuhmann. "Growing actin networks form lamellipodium and lamellum by self-assembly." *Biophysical journal* 95.12 (2008): 5508-5523.

Zimmermann, Juliane, et al. "Actin filament elasticity and retrograde flow shape the force-velocity relation of motile cells." *Biophysical journal* 102.2 (2012): 287-295.

3) They could add to the discussion a short comparison between their results and observations of the organization of the cortical actin in cells. This can then give strong motivation to study in the future more deformable vesicles.

Response to Referees

we have addressed all of the reviewer critiques in the revised version of our manuscript. We provide a point-by-point description of our modifications below.

Thank you very much,

Matt Peterson, Aparna Baskaran, Michael Hagan

Reviewer #1 (Remarks to the Author):

The authors have adressed my concerns. The Manuscript is now ready for publication.

Reviewer #2 (Remarks to the Author):

The authors have addressed my comments and I would fully support publication of this exciting manuscript in Nature Communications.

A minor comment: personally, I found log-log scale of Fig. 4 to be very impressive and far more revealing than the linear version presented in the manuscript. But I concede that this is to a certain degree a matter of preference.

Response: We went with the reviewer's suggestion, and put the log-log version into the main text. The one drawback of this version is that the range states, which correspond to 0 caps, cannot be presented on the log scale. Therefore, we have also put the linear scale version into the supplement.

Reviewer #3 (Remarks to the Author):

The authors have greatly improved the manuscript, and many questions regarding their observations are now more clearly explained. I recommend publication, but have these additional small comments:

1) Regarding my previous question about the stabilization of the rings along the rim of the oblate-shaped vesicle: can they calculate the balance of forces between the outwards radial push of the rotating filaments and the restoring force of the bent rim of the flattened vesicle ?

This oblate shape is similar in shape and balance of forces to the pancake-like shapes found in [35], using a different self-organization mechanism.

Response: This is a great suggestion, and we have followed it. We parameterize the vesicle as an ellipsoid, and estimate the elastic restoring force as a function of the major radius, in the limit of small deformations from the spherical shape. We equate this to the normal force exerted by the filaments in the polar band. The scaling form suggested by this theory agrees very well with data from the simulations, as shown in Supplementary Figure 6. We have also added a sentence to the results section that refers to this calculation and figure.

2) In the introduction they should explicitly mention the main example of cellular bio-polymer that affects the membrane shape, ie actin. In this context, they could explicitly mention that [12] found that actin filaments tend to "funnel" into a local deformation, as they find with their caps.

Also, there are several actin-specific simulations that seem relevant enough to mention, here are some that I found:

Ni, Haoran, and Garegin A. Papoian. "Membrane-median: Simulating deformable vesicles containing complex cytoskeletal networks." bioRxiv (2021).

Zhuravlev, Pavel I., Longhua Hu, and Garegin A. Papoian. "Computer Simulations of Mechano-Chemical Networks Choreographing Actin Dynamics in Cell Motility." Computational Modeling of Biological Systems. Springer, Boston, MA, 2012. 231-256.

Huber, Florian, Josef Käs, and Björn Stuhmann. "Growing actin networks form lamellipodium and lamellum by self-assembly." Biophysical journal 95.12 (2008): 5508-5523.

Zimmermann, Juliane, et al. "Actin filament elasticity and retrograde flow shape the force-velocity relation of motile cells." Biophysical journal 102.2 (2012): 287-295.

Response: Thank you very much for bringing this to our attention. We have now modified the introduction to include a discussion of these ideas and these citations.

3) They could add to the discussion a short comparison between their results and observations of the organization of the cortical actin in cells. This can then give strong motivation to study in the future more deformable vesicles.

Response: We have implemented this excellent suggestion in our modified discussion.